# Transcription Factor Movement and Exercise-Induced Mitochondrial Biogenesis in Human Skeletal Muscle: Current Knowledge and Future Perspectives

**DOI:** 10.3390/ijms23031517

**Published:** 2022-01-28

**Authors:** Dale F. Taylor, David J. Bishop

**Affiliations:** Institute for Health and Sport (iHeS), Footscray Park, Victoria University, Melbourne 8001, Australia; dale.taylor4@live.vu.edu.au

**Keywords:** mitochondrial biogenesis, exercise, skeletal muscle, transcription factors, subcellular

## Abstract

In response to exercise, the oxidative capacity of mitochondria within skeletal muscle increases through the coordinated expression of mitochondrial proteins in a process termed mitochondrial biogenesis. Controlling the expression of mitochondrial proteins are transcription factors—a group of proteins that regulate messenger RNA transcription from DNA in the nucleus and mitochondria. To fulfil other functions or to limit gene expression, transcription factors are often localised away from DNA to different subcellular compartments and undergo rapid movement or accumulation only when required. Although many transcription factors involved in exercise-induced mitochondrial biogenesis have been identified, numerous conflicting findings and gaps exist within our knowledge of their subcellular movement. This review aims to summarise and provide a critical analysis of the published literature regarding the exercise-induced movement of transcription factors involved in mitochondria biogenesis in skeletal muscle.

## 1. Introduction

### 1.1. The Importance of Skeletal Muscle in Human Health

Skeletal muscle comprises approximately 40% of an average human’s total mass [1]. The primary role of skeletal muscle is the conversion of chemical energy into mechanical movement, which can range from walking to world-class sporting performance. Critical for human health, reduced skeletal muscle mass through inactivity, disease-induced atrophy, or sarcopenia leads to physical disability, a lowered quality of life, and ultimately death [2]. A key feature of skeletal muscle is its high plasticity in response to homeostatic perturbations, such as those induced by hormones, nutrition, or exercise. To adapt to changing energy needs, skeletal muscle relies on the mitochondrion—a structure commonly known as the powerhouse of the cell [3,4].

### 1.2. Skeletal Muscle and the Mitochondria

Engulfed by an early eukaryote over a billion years ago, the mitochondrion is a double membrane bound organelle that has become a vital component of human life (Figure 1) [5]. Mitochondria facilitate various homeostatic processes, including apoptosis, calcium ion cycling, and energy generation [6]. Discrete compartments within the mitochondria are used to achieve these diverse functions, with an outer mitochondrial membrane (OMM) separating the contents of the cytosol from the intermembrane space (IMS). An inner mitochondrial membrane (IMM) forms the folded cristae—a structure important for energy generation that also separates the IMS from the encapsulated interior matrix [5]. Small hydrophilic molecules are able to enter the IMS through the voltage-dependent anion channel (VDAC) [7]. Larger molecules and proteins can move into the IMS through association with the translocase of the outer membrane (TOM) complex and into the matrix through the translocase of the inner membrane (TIM) complex and their related factors [7]. Due to its origin as proteobacteria, multiple copies of circular mitochondrial DNA (mtDNA) are retained in the matrix [8]. Encoding 13 proteins essential for its function, translation of genes present in mtDNA occurs by ribosomes in the mitochondrial matrix. To facilitate protein synthesis, two of the ribosomal RNAs (rRNA) that form part of mitochondrial ribosomes and 22 transfer RNAs (tRNA) are also encoded within the mtDNA [8].

Unlike the nucleus, which is located only on the periphery of skeletal muscle cells, otherwise known as myocytes or muscle fibres, mitochondria exist as two morphologically distinct populations within skeletal muscle (Figure 1) [9,10]. Located beneath the myocyte plasma membrane, also termed the sarcolemma, are subsarcolemmal (SS) mitochondria. Large and oval shaped, SS mitochondria comprise approximately 10% of the total mitochondrial volume in muscle [10]. Intermyofibrillar mitochondria (IMF), which comprise the remaining 90% of the total mitochondrial volume, are smaller and capable of compacting between myofibrils—the contractile filaments composed of the motor proteins actin and myosin [10]. Although both mitochondrial populations are discrete, mapping of mitochondrial membranes indicates some continuity between the two through membrane linkages [11]. Aside from morphological differences, IMF mitochondria exhibit a higher capacity for energy generation than SS mitochondria, which is achieved, in part, through use of an extensive interconnected reticulum network [10,11]. While not as crucial for providing energy to myofibrils, SS mitochondria still play a vital role in exercise by heavily regulating calcium homeostasis and therefore muscle contraction [10].

## 2. Mitochondrial Biogenesis and the Role of Transcription Factors

As a result of their bacterial origin, mitochondria cannot be created de novo and are instead made through mitochondrial biogenesis—a process most commonly defined as the synthesis of new components of the mitochondrial reticulum [12]. The expression of genes (stretches of DNA that code for proteins) is controlled by transcriptional regulators—of which the largest group are transcription factors (Figure 2). Transcription factors are proteins that bind to DNA and interact with the transcriptional machinery to either induce or repress messenger RNA (mRNA) transcription [13]. Proteins that control transcription factor activity but lack the ability to bind to DNA themselves are called transcriptional co-activators and co-repressors and are often conflated with transcription factors as one group [13,14]. The access of transcription factors to specific DNA sequences is partly controlled by chromatin structure, with remodelling through histone compaction decreasing rates of transcription [15]. Transcription factors often regulate multiple unrelated genes and even carry out additional roles distinct from transcription throughout the cell [13]. To tightly control transcriptional activity, transcription factors are often localised to different subcellular compartments of the cell [16].

Changes in transcription factor abundance in different cellular compartments, such as the nucleus and mitochondria, are initiated by various signalling networks that detect homeostatic perturbations [17]. These changes are rapid and can be triggered in skeletal muscle by a single session of exercise [18,19]. While many studies have investigated the effects of exercise on gene expression in skeletal muscle, changes in the abundance of transcription factors in different cellular compartments that dictate these increases have received limited attention [20]. This review aims to summarise the current knowledge and provide a critical analysis of the exercise-induced movement of transcription factors involved in mitochondrial biogenesis in human skeletal muscle.

## 3. Regulation of Transcription Factor Abundance throughout the Cell

### 3.1. Movement across Intracellular Membranes

To control gene expression, transcription factors and their associated regulators can move into subcellular compartments through different mechanisms depending on the size, type, and charge of the molecule (Figure 3). Many proteins, including transcription factors, require co-factors to bind for their function [21,22]. These co-factors are typically smaller molecules, such as calcium ions, lipids, and various metabolites, which are often capable of passive diffusion through phospholipid membranes [23,24]. However, they may also be actively transported using ion channels and transporters [24,25]. For the bulk movement of molecules intracellular transport vesicles are often utilised, particularly from the endoplasmic reticulum and Golgi apparatus [26]. This mechanism is also used for integrating transmembrane proteins, the protruding domains of which can subsequently be cleaved to further facilitate translocation [27].

Passive diffusion in and out of the nucleus by metabolites and smaller proteins (<50 kDa) is possible through the nuclear pore complexes (NPC), large multimeric structures embedded in the double membrane [28]. The active movement of proteins across the nuclear membrane is facilitated through the binding of protein chaperones that are recognised by the NPC [28]. To localise to the mitochondria, unfolded proteins bind to a combination of heat shock proteins (HSP) 60/10 and HSP70 chaperones [29]. These unfolded proteins are subsequently able to pass through the TOM and TIM complexes in the OMM and IMM respectively [29]. Small hydrophilic metabolites can passively diffuse in and out of the IMS through VDAC, similarly to the NPC for the nucleus [30,31].

### 3.2. Targeting of Transcription Factors to Specific Compartments

The localisation of transcription factors to different compartments of the cell is one mechanism by which their activity is regulated [16]. To localise to different subcellular compartments, proteins rely on short amino acid sequences in their peptide structure called localisation sequences that are recognised by protein chaperones [32]. However, they are often hard to identify solely in silico from amino acid sequence and must be validated experimentally [32]. Proteins may have a single localisation sequence to one compartment, an ambiguous signal that can allow localisation to multiple compartments, or multiple localisation sequences to different subcellular compartments (Figure 4) [33].

Complicating the prediction of localisation based on amino acid sequence, is the ability of protein folding to conceal or expose localisation sequences [33]. Protein movement may also be impeded by the binding of additional proteins that obscure access of the localisation sequences to import and export machinery [33]. Post-translational modifications, such as phosphorylation and acetylation, may also modulate localisation sequences by affecting both protein folding and the ability for chaperone and inhibitory proteins to bind [34].

The accumulation of a transcription factor in a specific subcellular compartment may not always indicate increased translocation but could instead be due to an increase in production or a reduction in degradation [35]. An example of the later mechanism not involving transcription factors is that of Pink1/Parkin—proteins that form part of a mitochondrial quality control system [36]. Under basal conditions Pink1 continually passes through the TOM but is subsequently cleaved in the IMS leading to degradation by proteasomes and no accumulation; however, upon mitochondrial damage Pink1 accumulates by tightly binding to the TOM and is able to recruit cytosolic Parkin to trigger mitophagy [36]. Therefore, an increase in a specific protein, such as transcription factors, within an organelle may not always indicate movement from a pre-existing population to another compartment [37].

Complicating the prediction of localisation based on amino acid sequence, is the ability of protein folding to conceal or expose localisation sequences [33]. Protein movement may also be impeded by the binding of additional proteins that obscure access of the localisation sequences to import and export machinery [33]. Post-translational modifications, such as phosphorylation and acetylation, modulate localisation sequences by affecting both protein folding and the ability for chaperone and inhibitory proteins to bind [34]. Several different techniques are available to researchers to identify the potential movement or accumulation of proteins within a specific subcellular compartment.

## 4. Measuring Transcription Factor Movement Experimentally

To measure the movement of transcription factors in skeletal muscle numerous techniques have been used, each with their own advantages and disadvantages.

### 4.1. Subcellular Fraction of Skeletal Muscle

Although various methods currently exist for the analysis of proteins, observations of movement between subcellular compartments are commonly obscured by using whole-muscle samples. Using whole-muscle samples is problematic for analysing protein movement as there is currently no method for analysing proteins that move between compartments but do not change in total abundance. Although protocols exist for deriving crude nuclear, cytosolic, and mitochondrial fractions, the remnants of other fractions are likely to impair the detection of significant changes (Figure 5A) [38,39,40]. Despite this limitation, the analysis of different subcellular fractions has been performed using various qualitative and quantitative techniques to gain insight into exercise-induced changes to transcription factors involved in mitochondrial biogenesis in skeletal muscle.

### 4.2. Immunodetection

The most-used technique to investigate the possible movement of transcription factors in response to exercise has been immunodetection via western blotting, histochemistry, and enzyme-linked immunosorbent assays (ELISA). The largest drawback of this approach is the reliance on selecting a known target, making it liable to miss changes to proteins that have not yet been identified or characterised. The statistical significance of the observed changes is also difficult to assess, as is the comparison between different proteins, with antibodies possessing different binding affinities to both the target protein and the conjugated antibodies that are used to generate the detectable luminescence [41]. Often antibodies will not just target the desired protein but also other proteins that contain the same, or similar, amino acid epitope [42]. Validation of antibody efficacy through a knockout or overexpression cell model may be required to ensure the detected band is the target protein [43]. This is pertinent for transcription factors involved in mitochondrial biogenesis, with many possessing multiple different isoforms that have varying sizes and functions [44]. The activity state of the protein, based on modifications such as phosphorylation and acetylation, can also often not be accurately discerned based on antibody detection. As a consequence, the accumulation of a transcription factor in a subcellular compartment may not be reflective of an increase in activity.

### 4.3. Mass Spectrometry-Based Proteomics

Biochemical techniques based on analytical computation, such as mass spectrometry, are increasingly being used for analysing proteins in skeletal muscle after exercise (Figure 5B). The main benefit of mass spectrometry over traditional proteomic techniques is that it is non-targeted and therefore capable of detecting all proteins within a sample. This has led to the creation of large databases that can be used to identify unknown proteins within a sample [46]. Although mass spectrometry can detect any protein within a sample, the presence of large contractile proteins within muscle significantly impedes the identification of low abundance proteins such as transcription factors [45]. Furthermore, because of the need to generate cleaved peptides through proteases such as trypsin, the use of protease inhibitors is constrained and leads to the degradation of proteins during sample preparation [47]. This is particularly problematic for detecting transcription factors due to their low abundance [48]. However, strategies do exist for improving the number of protein detections through the separation of one sample into several by the varying physicochemical properties of the proteins, such as size [49]. Additionally, transcription factors may be isolated from other proteins through affinity-based techniques targeting DNA; however, this may exclude transcription factors that only bind to DNA under specific conditions [50]. The subcellular fractionation of tissue has successful been applied to mass spectrometry-based proteomics, with fractions significantly enriched with proteins from the target compartment [51,52,53]. A large advantage of analysing subcellular fractions using mass spectrometry is the validation of purity by comparing the set of detected proteins to databases of proteins that are annotated with an experimentally validated subcellular localisation [54]. Although mass spectrometry has become more accessible, no study has used this technique to investigate the changes of transcription factors in skeletal muscle subcellular fractions, such as the nucleus and mitochondria, in the context of exercise-induced mitochondrial biogenesis.

### 4.4. Electrical Stimulated Contraction and Microscopy

To directly analyse the movement of proteins following contraction, confocal microscopy has been employed within various skeletal muscle systems [55,56,57]. To detect a specific protein, a fluorescent tag is inserted into the coding region of the gene leading to the expression of a protein containing a conjugated tag that is detectable via absorption and emission of specific wavelengths of light [58]. However, correctly inserting a fluorescent tag is difficult due to tertiary structure potentially obscuring the construct from light exposure [58]. Additionally, fluorescent tags can often bind together and aggregate leading to incorrect localisation and cellular toxicity [59]. The main drawback of this system is the inability of it to be utilised within in vivo muscle, with imaging only possible in ex vivo and in vitro skeletal muscle systems that contract through electrical stimulation [55,60]. While visible myotubule shortening occurs with electrical stimulation, it is debatable whether this process is equivalent to that of contraction within in vivo tissue, with several important biomarkers of mitochondrial biogenesis unchanged [61]. Comparisons between in vitro and in vivo cells is also problematic due to the process of immortalisation often modifying the regulation of protein expression and activity [62].

### 4.5. Additional Issues

Further limitations of these techniques include the lack of differentiation between SS and IMF mitochondria, with differing content and function likely influencing transcription factor localisation [10,63]. The detection of transcription factor translocation may also be constrained by the small number and timing of muscle biopsy samples taken post exercise in most studies [64]. This has led to an incomplete picture of the post-exercise movement of proteins that contribute to mitochondrial biogenesis, and has likely prevented the detection of some transcription factor changes that occur at times not covered by typical biopsy times. Although these technical shortcomings have limited our understanding, evidence has been provided to support the exercise-induced movement of numerous transcription factors between subcellular compartments to regulate mitochondrial biogenesis in skeletal muscle.

## 5. Exercise-Induced Movement of Transcription Factors in Skeletal Muscle

### 5.1. Additional Contraction and Homeostatic Perturbations

Exercise-induced movement of transcription factors begins with various homeostatic perturbations triggered by contraction in skeletal muscle (Figure 6). For a detailed explanation of all the homeostatic perturbations that occur in skeletal muscle during exercise the reader is encouraged to read Hawley et al. (2018) [65]. A brief summary of some of the key changes related to exercise-induced mitochondrial biogenesis is described below.

Muscle contraction is initiated by an action potential from a motor neuron, which elicits sarcolemma depolarisation and stimulates the ryanodine receptor 1 (RyR1) located on the internal sarcoplasmic reticulum (SR) to release calcium ions into the cytosol [66]. As exercise begins, the small amount of basal intramuscular ATP is hydrolysed to fuel contraction and results in an increase in the ratio of ADP to ATP throughout the cell [65]. Additionally, the amount of free cellular AMP increases as adenylate kinase catalyses the formation of both an ATP and an AMP molecule from 2 ADP molecules to quickly increase ATP levels [65].

During the early stages of exercise, ADP is also rapidly converted back into useable ATP through the transfer of a phosphate from stores of intramuscular phosphocreatine with a resultant increase in free creatine [65]. Muscle contraction also stimulates glycogenolysis and glycolysis (the breakdown of muscle glycogen and glucose respectively), with an increase in lactate concentration that depends on the balance between glycolytic rate and subsequent mitochondrial metabolism of pyruvate and lactate [67]. The longer exercise continues the further the oxidative capacity of mitochondria is employed for ATP generation, with the ratio of NAD^+^ to NADH increasing as a by-product of mitochondrial respiration [68]. Although the mitochondrial membranes are impermeable to both NAD^+^ and NADH by passive diffusion, they are shuttled across the IMM by the mitochondrial nicotinamide adenine dinucleotide transporter (MCART1) [69]. The increase in mitochondrial respiration also serves to lower the partial pressure of oxygen (pO_2_) within skeletal muscle, inducing acute localised hypoxia [70].

While always present at physiological levels in muscle, reactive oxygen species (ROS) accumulate significantly upon contraction [71,72]. The formation of ROS was initially thought to occur predominantly by increased oxygen consumption during respiration, with an exponential relationship between decreased pO_2_ and increased ROS identified [73]. However, ROS likely arise through many different sources during exercise, including from metabolic processes outside the mitochondria [74]. Traditionally seen as a means to quickly regenerate ATP during exercise, lactate also contributes to increasing ROS production through raising the rate of aerobic respiration and through interactions with iron in the mitochondria [75,76,77]. As a consequence of their size, ROS and their by-products, such as hydrogen peroxide, are able to freely diffuse into different compartments and influence transcription factor activity [73].

Another class of molecules that diffuse throughout all compartments of the cell are free fatty acids (FFA)—the concentration of which increase, particularly for use in beta-oxidation, during submaximal exercise [78,79]. Outside of skeletal muscle, exercise stimulates the release of various molecules (e.g., adrenaline, insulin-like growth factor, and cytokines) into the blood stream that can bind to receptors on the extracellular side of the sarcolemma and elicit internal signalling [80,81,82,83]. In particular, the binding of adrenaline from the blood stream to β-adrenergic receptors on the sarcolemma of muscle stimulates adenylate cyclase to catalyse the reaction of intracellular ATP to cyclic AMP (cAMP) [84]. Additional cAMP is also generated by soluble adenylate cyclase within the mitochondrial matrix, a reaction that is activated by HCO_3_^−^ derived from CO_2_ released by the tricarboxylic acid (TCA) cycle [85,86].

The aforementioned perturbations are detectable by the various signalling networks present within skeletal muscle, which in turn regulate the activation of transcription factors to stimulate mitochondrial biogenesis.

### 5.2. Calmodulin-Dependent Kinase II (CaMKII)

The homeostatic perturbations induced by exercise stimulate various signalling pathways that affect both the activity and localisation of transcription factors in skeletal muscle (Figure 7). One of the most studied regulators of transcription factors is calmodulin-dependent kinase II (CaMKII)—a protein activated by cytosolic calcium binding to calmodulin (CaM), which then binds to CaMKII causing autophosphorylation and activation [87]. A single session of exercise significantly increases the activity of CaMKII in human skeletal muscle, with an intensity-dependent activation observed [88,89]. Analysis of CaMKII has primarily relied on immunoblotting of homogenised whole-muscle lysate, with antibodies capable of differentiating between inactive and active protein through phosphorylation on the threonine at amino acid residue 286 (Thr^286^) or amino acid residue 287 (Thr^287^) depending on the specific CaMKII isoform [88]. A large amount of research on the localisation of CaMK following exercise has focused on calmodulin-dependent kinase IV (CaMKIV) using animal models. Unfortunately, CaMKIV is not highly expressed within human skeletal muscle and likely acts on different substrates than CaMKII [90,91]. In human skeletal muscle, CaMKII predominantly exists within the cytosol; however, its activity is increased within the nucleus following exercise, as determined by subcellular fractionation with immunoblotting and microscopy [92,93,94]. This may be via increased movement into the nucleus as several isoforms of CaMKII contain a nuclear localisation sequence (NLS)—the inhibition of which may occur through the assembly of individual subunits into the larger functional enzyme, or multimer, in the cytosol [95,96]. Additionally, importation of assembled CaMKII is likely further regulated by phosphorylation of residues in the NLS [95,96]. However, no research to date has been able to confirm if CaMKII does accumulate within the nucleus following a single session of exercise [87].

Further activation of CaMKII occurs by peroxisome proliferator-activated peceptor γ coactivator 1- and estrogen-related receptor-induced regulator in muscle 1 (PERM1)—a skeletal muscle specific regulator that has increased protein expression following a single session of exercise in humans [97,98]. While the mechanism of PERM1 function is unclear, overexpression in mouse skeletal muscle increases several markers of mitochondrial biogenesis—including mtDNA content and citrate synthase activity [97]. Based on the analysis of conserved amino acid residues, PERM1 has both nuclear import and export sequences indicating localisation may play a role in regulating its function [99]. Subcellular fractionation and immunoblot analysis of in vitro mice myotubules reveals PERM1 is predominantly localised within the cytosol, and to a lesser extent the nucleus, similar to that of CaMKII [97,98]. However, no research has explored PERM1 localisation within human skeletal muscle or the effect of exercise on PERM1 localisation.

To exert its primary effect on mitochondria biogenesis, CaMKII contributes to the phosphorylation of the transcription factor cyclic adenosine monophosphate response element binding protein (CREB) [92,100]. Immunoblot analysis of homogenised human skeletal whole-muscle lysate indicates no change in total CREB protein content, but an increase in active CREB via phosphorylation of serine 133 (Ser^133^) following a single session of exercise [92]. An increased phosphorylation of nuclear CREB has been detected following exercise, with an amplified binding of CREB to the gene promotor of peroxisome proliferator-activated receptor gamma coactivator 1-alpha (PGC-1α)—a protein commonly called the ‘master regulator’ of mitochondrial biogenesis [88,100,101,102]. Additionally, CREB may be capable of importation into the mitochondria, where it is thought to be critical for the transcription of genes encoded within mtDNA [103]. However, this has only been demonstrated within murine neuronal and liver cells and may not be applicable to human skeletal muscle [101,103]. While predominantly thought of as located within the nucleus, it is unclear what percentage of CREB is localised throughout the different subcellular compartments in human skeletal muscle following exercise.

### 5.3. Peroxisome Proliferator-Activated Receptor Gamma Coactivator 1-Alpha (PGC-1α)

No transcription factor is more researched within the field of mitochondrial biogenesis than PGC-1α, an intrinsically disordered protein (IDP). The major protein binding domain of PGC-1α is highly variable and this enables it to bind to many different targets through changes in tertiary structure [104]. Within mitochondrial biogenesis, PGC-1α acts as the central regulator transferring the transient signalling of calcium fluctuations, energy stress, and changes in metabolites to stable protein expression [105]. While lacking DNA binding capabilities itself, PGC-1α acts as a co-factor for numerous transcription factors, including the nuclear respiratory factor (NRF) and peroxisome proliferator-activated receptor (PPAR) families [102].

Evidence of PGC-1α’s central role in exercise-induced mitochondrial biogenesis comes partly from overexpression of the PGC-1α protein in mice models and ex vivo human myotubules, with a consistent increase in other mitochondrial biogenesis regulators, mitochondrial proteins, and mtDNA copy number observed [14,106,107]. However, contrary to expectations, mice with whole-body PGC-1α gene inactivation are able to increase their mitochondrial content with exercise training to similar levels as the wild type variant [108,109]. This would suggest there are other pathways or mechanisms that induce mitochondrial biogenesis in skeletal muscle following exercise. However, several aberrant side effects in PGC-1α whole-body knockout mice were detected, such as increased activity of mitochondrial biogenesis regulators even while at rest compared to wild-type mice [110]. The importance of PGC-1α has also been investigated through muscle-specific gene inactivation in mice. These mice present with a decreased tolerance to exercise and a reduction in mitochondrial gene expression, consistent with a central role of PGC-1α in mitochondrial biogenesis [111]. Previous studies into PGC-1α function are often limited due to not analysing the expression or activity of other peroxisome proliferator-activated receptor gamma coactivator family members. In particular, there has been little research on peroxisome proliferator-activated receptor gamma coactivator 1-beta (PGC-1β)—a poorly understood protein with potentially overlapping function and therefore movement as PGC-1α [108,109,112,113]. Whether or not PGC-1β undergoes translocation like that of PGC-1α following exercise is unknown and warrants investigation.

At rest, PGC-1α can be identified at very low levels in both the nucleus and mitochondria; however, it is primarily detected within the cytosol of skeletal muscle [114,115]. After exercise PGC-1α is activated through deacetylation and phosphorylation, which prevents ubiquitin-mediated proteolysis and leads to an accumulation in the nucleus [114,115,116]. Initial investigation of PGC-1α using truncated proteins in vitro validates a nuclear localisation, with a NLS in the C-terminus region of the full length isoform [117]. The translocation of PGC-1α to the nucleus is further supported by the immunoblotting of separated nuclear fractions following exercise [18,114,116,118,119]. Once inside the nucleus, PGC1-α can bind to other transcription factors leading to an increase in transcription [14,102]. As part of a positive feedback loop, PGC-1α increases its own expression by binding to the transcription factor family myocyte-specific enhancer factor 2 (MEF2) at its own promoter region [120,121]. The rapid expression of PGC-1α mRNA occurs following exercise, with RNA sequencing (RNA-Seq) and quantitative PCR (qPCR) finding a 5-fold increase approximately 2 to 5 h after the end of exercise in skeletal muscle [20,122,123]. This ultimately leads to more PGC-1α protein expression and further accumulation in the nucleus.

In addition to accumulating within the nucleus, PGC-1α may also localise to the mitochondria in skeletal muscle following a single session of exercise [63]. However, evidence of this phenomenon is limited, with one study using immunoblotting of subcellular fractions in rat skeletal muscle finding PGC-1α to localise only in SS mitochondria [63]. This may in part explain why increases in SS mitochondria volume are greater than IMF mitochondria in response to exercise; however, more evidence is required to support this hypothesis (44). Conversely, one study using subcellular fractionation and immunoblotting on human skeletal muscle showed no significant change in mitochondrial PGC-1α protein content, both immediately after or 3 h post exercise [37]. The exact role PGC-1α may fulfil within the mitochondria of skeletal muscle is unclear. One study in human carcinoma cells reported PGC-1α to localise with mitochondrial transcription factor A (TFAM), a non-sequence specific transcription factor required for mtDNA expression and replication [124,125]. However, this study relied on confocal microscopy and not a more direct method of analysing protein-protein interaction such as co-immunoprecipitation [124].

Although most studies investigating the role of PGC-1α in mitochondrial biogenesis only focus on the canonical 797 amino acid PGC-1α protein, various other isoforms have been detected in human skeletal muscle [44,126]. It is important to consider all variants of PGC-1α in skeletal muscle, due to potential overlaps in function and research identifying an exercise intensity-dependent expression of different isoforms [127]. One noteworthy group of isoforms is splice variant N-terminal (NT)-PGC-1α, a group of truncated isoforms characterised by an in-frame stop codon that results in proteins of approximately 270 amino acids [128]. Analysis using chromatin immunoprecipitation sequencing (ChIP-seq) reveals that PGC-1α and NT-PGC-1α likely carry out largely overlapping transcriptional co-factor functions [129]. Due to the lack of a C-terminal region, NT-PGC-1α lacks the canonical NLS of the full length PGC-1α [128,130]. While the mechanism of NT-PGC-1α translocation to the nucleus is unclear, accumulation is partially controlled by an exposed nuclear export sequence created by the premature stop codon [130]. An increase in the levels of nuclear cAMP, which occurs during exercise, leads to the phosphorylation of NT-PGC-1α residues that prevent nuclear exportin 1 (XPO1) mediated export to the cytosol, resulting in accumulation in the nucleus [128,130]. The mRNA expression of NT-PGC-1α has been shown to increase 2 h post both resistance and endurance exercise, similar to the full length PGC-1α isoform [131]. However, the possible nuclear accumulation of NT-PGC-1α protein in human skeletal muscle following exercise has not been investigated, with previous work investigating PGC-1α largely ignoring NT-PGC-1α.

The nuclear respiratory factor 1 (NRF-1) is a transcription factor that serves to induce expression of various proteins involved in oxidative phosphorylation (OXPHOS), the last stage of aerobic respiration, and numerous other mitochondrial genes in the nucleus [14,132]. Although NRF-1 is only detected within the nucleus, its activity is controlled by proteins that move between subcellular compartments [133]. Aside from PGC-1α, NRF-1 function is regulated by the binding of nuclear factor erythroid 2-like 2 (NFE2L2)—a transcription factor that under basal conditions is detected in the cytosol of in vitro human cells and mouse skeletal muscle [133]. An increase in cytosolic ROS and AMP during exercise leads AMP-activated protein kinase (AMPK) to phosphorylate kelch ECH associated protein 1 (KEAP1), a protein attached to NFE2L2 [133]. Phosphorylation of KEAP1 leads to its detachment from NFE2L2, exposing a NLS on the latter that causes it to accumulate in the nucleus [134,135,136]. Knockdown of NFE2L2 protein expression in mice highlights the importance of its regulation on NRF1, with a resultant exercise intolerance and reduced markers of mitochondrial biogenesis within mouse skeletal muscle [137,138]. As with most proteins, movement of NFE2L2 to the nucleus from the cytosol following exercise has yet to be demonstrated within human skeletal muscle.

The PPAR family of transcription factors (α, β/δ, γ) are responsible for the expression of various mitochondrial proteins, including genes involved in fatty acid oxidation [139]. Although PPAR members are predominantly nuclear, a major regulatory mechanism for their activation is localisation—both between organelles and within compartments inside the nucleus [140,141]. During exercise, the binding of small endogenous fatty acids was shown to prevent ubiquitin-mediated proteolysis of inactive perinuclear PPARα and PPARβ/δ [142,143,144]. Due to the binding of ligands, PPARα and PPARβ/δ are then capable of undergoing heterodimerisation with retinoid X receptor (RXR), allowing subsequent intranuclear localisation to promoter regions where they can bind with PGC-1α and to DNA [141,145]. Inactive PPARα has also been detected in the cytoplasm, with shuttling to the nucleus controlled by the binding of centrosome-associated protein CAP350, the concentration of cytosolic calcium, and the presence of ligands [140,146]. Currently, nucleo-cytoplasmic shuttling of PPARα has only been demonstrated within in vitro human fibroblasts, and whether exercise initiates increased nuclear localisation in skeletal muscle is unknown.

### 5.4. Calcineurin (CaN)

In addition to the activation of CaMKII, the binding of calcium to calmodulin allows the activation of the phosphatase CaM-dependent calcineurin (CaN) [21]. Through dephosphorylation, CaN facilitates the translocation of transcription factors such as nuclear factor of activated T-cells (NFAT), transcription factor EB (TFEB), and CREB regulated transcription coactivator (CRTC) members from the cytoplasm to the nucleus [147,148,149]. The NFAT family of proteins are transcription factors that are implicated heavily in both the development and maintenance of skeletal muscle [150]. The interaction of CaN with NFAT has been demonstrated extensively using in vitro models, with transgenic overexpression of CaN activity in mice myotubules significantly increasing NFAT localisation to the nucleus [151]. Evidence of NFAT translocation due to exercise comes both from confocal microscopy of electrically stimulated ex vivo mouse skeletal muscle, as well as immunoblotting in human skeletal muscle [147,152]. Different intensities of exercise result in varying NFAT members accumulating in the nucleus and thus may form part of the mechanism that stimulates intensity-dependent mitochondrial adaptation in skeletal muscle [153,154,155].

The TFEB protein is primarily involved in expressing genes for lysosome and autophagosome formation, organelles critical to mitochondrial homeostasis through metabolite exchange and targeted degradation of dysfunctional components [156,157]. Although most evidence of TFEB translocation comes from various in vitro and in vivo human starvation models, exercise has been shown to increase its nuclear abundance in skeletal muscle [152,158,159]. Outside of its role in lysosomal regulation, TFEB also induces mitochondrial biogenesis by directly increasing expression of PGC-1α mRNA in the nucleus [160]. Mice overexpressing TFEB in skeletal muscle were found to have 38 genes involved in mitochondrial biogenesis and function upregulated, while mice with a muscle-specific knockout had 73 genes downregulated [161]. Interestingly, TFEB appears to induce mitochondrial biogenesis outside of increasing PGC1α activity, with PGC1α knockout mice increasing mitochondrial content without PGC1β compensation through TFEB overexpression [161]. This indicates there are likely other regulators that TFEB interacts with to stimulate exercise-induced mitochondrial biogenesis outside of the canonical PGC-1α pathway.

Regulation of CREB activity is controlled by the CRTC family, a group of transcription factors that has three members—CRTC1, CRTC2, and CRTC3 [162]. Unlike CRTC1, which is expressed only in the brain, both CRTC2 and CRTC3 are highly expressed in skeletal muscle [163]. During exercise, the activation of CaN leads to the dephosphorylation of CRTC2 and CRTC3, with a subsequent accumulation in the nucleus occurring in human skeletal muscle [162,164,165]. Overexpression of CRTC2 in mice leads to increased CREB activity and PGC-1α expression in skeletal muscle, highlighting its role in mitochondrial biogenesis [164]. Interestingly, only CRTC3-CREB conjugates were detected bound to the promoter of PGC-1α in mouse liver hepatocytes after the addition of rotenone, an inhibitor of OXPHOS that stimulates a deprivation of ATP similar to maximal exercise [166]. Therefore, CRTC2 and CRTC3 likely have non-redundant functions, with conditions such as the intensity or duration of exercise potentially affecting their activity in skeletal muscle.

### 5.5. AMP-Activated Protein Kinase (AMPK)

Connecting a need for increased ATP generation to a transcriptional response is the energy sensor AMPK [22]. Activation of AMPK in skeletal muscle occurs rapidly via phosphorylation after the start of exercise when the transient level of ATP is at its lowest and the level of free AMP is highest [167]. The activity of AMPK is further upregulated by the ROS nitric oxide (NO^−^), a relationship that has been demonstrated by immunoblotting of in vitro rat myotubules [168,169]. These cellular perturbations cause AMPK to be phosphorylated and to subsequently accumulate within the nucleus following exercise, as determined by immunoblotting using fractionated human skeletal muscle [170]. However, only heterotrimeric AMPK that contains an alpha subunit 2 isoform is capable of this localisation, with the other isoform, alpha subunit 1, not possessing a NLS and instead phosphorylating only cytosolic targets [171,172,173].

Control of transcription by nuclear AMPK occurs through phosphorylation of epigenetic regulators, such as the transcriptional repressors histone deacetylase 4 (HDAC4) and histone deacetylase 5 (HDAC5) [173,174,175]. The HDAC proteins modify chromatin structure by deacetylating histones, the complexes that DNA binds around, leading to a condensed structure and preventing transcription factor access [176]. The phosphorylation of HDAC complexes leads to their subsequent translocation into the cytosol where they can interact with other proteins or be degraded by proteolysis [173,174,175]. This interaction has been demonstrated by immunoblotting of in vivo mouse and rat myotubules and human skeletal muscle that has been fractionated [173,177,178]. Regarding mitochondrial biogenesis, the removal of HDAC4 and HDAC5 from the nucleus allows access for MEF2 to bind on the upstream promotor of PGC-1α and induce its expression [121]. Additionally, AMPK also activates PGC-1α directly by phosphorylating several residues required for its transcriptional function as determined using immunoblotting of both in vivo and in vitro mouse skeletal muscle [179]. Whether AMPK completes this function within both the nucleus and cytosol, or within human skeletal muscle, requires further investigation. Evidence of cytosolic phosphorylation can be found in AMPK-deficient mice, with an inability to stimulate exercise-induced translocation of PGC-1α to the mitochondria from the cytosol [63].

Further modulation of PGC-1α activity occurs by AMPK increasing mitochondrial fatty acid oxidation, which increases the ratio of NAD^+^ to NADH and activates by phosphorylation the NAD^+^-dependent deacetylase sirtuin-1 (SIRT1) [180,181]. In response to exercise, SIRT1 activity increases within the nucleus where it can directly de-acetylate targets such as PGC-1α [182]. However, immunoblotting of in vivo mice samples and human muscle biopsies indicates SIRT1 protein content does not increase in the nucleus following exercise [182]. While SIRT1 is predominantly located in the nucleus, a cytosolic pool does exist [68]. However, it is not clear whether SIRT1 de-acetylates cytosolic PGC-1α, with most of its function in the cytosol likely related to apoptotic signalling and not mitochondrial biogenesis [182,183]. Localisation of SIRT1 to the mitochondria following exercise may also occur, with one study in a human carcinoma cell line finding SIRT1 co-localised with PGC-1α and TFAM in the inner matrix [124]. Whether it is performing the same role in the mitochondria as it does within the nucleus, or if this occurs in skeletal muscle, is unclear [184].

### 5.6. Tumour Protein P53 (p53)

The p53 protein is primarily involved in genome stability but is also increasingly being investigated for its ability to regulate transcription factors and mitochondrial biogenesis [37,185,186,187,188]. The localisation and activity of p53 appears to be altered by phosphorylation from AMPK and p38 mitogen-activated protein kinase (p38 MAPK), as well as deacetylation by SIRT1 [185,189,190]. Although continuously imported into the nucleus, under normal conditions p53 is rapidly translocated to the cytosol and ubiquitinated for degradation [191]. However, during periods of cellular stress, post-translational modifications to p53, such as phosphorylation, prevent cytosolic exportation and thus degradation [191]. This is reflected by a single session of exercise increasing p53 nuclear content as detected by immunoblotting within both human and rodent skeletal muscle immediately after, and 3 h post, exercise [37,190,192,193,194].

Of the known transcription factors involved in mitochondrial biogenesis, p53 is able to bind to phosphorylated CREB and subsequently CREB binding protein (CBP)—a large co-activator complex involved in the majority of cellular processes [195]. Additionally, p53 has been confirmed to bind to the nuclear transcription factor estrogen-related receptor alpha (ERRα) in human embryonic kidney cells [196]. This finding is significant due to ERRα regulating the expression of various OXPHOS and fatty acid oxidation genes, as well as TFAM [197,198]. Whether this interaction of p53 and ERRα increases post exercise within human skeletal muscle warrants investigation.

In the skeletal muscle mitochondria, accumulation of p53 after exercise is uncertain with both increases and decreases observed in mice and no change in humans [37,186,193]. It is therefore unclear whether p53 localising to the mitochondria is required for exercise-induced mitochondrial adaptations. Hypotheses for the role of p53 in mitochondria include providing genomic stability to mtDNA and increasing the transcription of mtDNA-encoded mitochondrial genes through interactions with TFAM [186,199].

### 5.7. Protein Kinase A (PKA)

The presence of increased levels of cAMP during exercise activates protein kinase A (PKA), a protein with various localisations in skeletal muscle [200]. During rest, the majority of PKA is tethered internally to the sarcolemma, with high levels of intracellular cAMP during exercise leading to the detachment of the catalytic subunit [200,201]. Once detached, the catalytic subunit can move into both the cytosol and nucleus [200,201]. Additional PKA remains tethered cytosolically on the outer mitochondrial membrane by A-kinase anchoring proteins (AKAPs), and as free populations inside the mitochondrial matrix [202,203]. Although the OMM is permeable to cAMP from the cytosol, it cannot freely diffuse into the mitochondrial matrix and is instead imported by a currently unknown mechanism [204,205]. While cAMP can be produced within the matrix, it is unclear if PKA phosphorylates CREB in the mitochondria—its main target within the nucleus for inducing mitochondrial biogenesis [206].

In addition, PKA has been demonstrated to phosphorylate NT-PGC1-α to prevent its exportation to the cytosol, increasing its nuclear accumulation [130]. Further upregulation of mitochondrial biogenesis occurs by PKA displacing FK506-binding protein 12 (FKBP12) from RyR1, the former stabilising the closed receptor conformation [207,208,209]. The phosphorylation of FKBP12 by both PKA and also CaMKII leads to a partially open conformation of RyR1, releasing additional calcium ions from the SR [207,208,209]. Evidence of PKA and CaMKII interactions with FKBP12 have been well studied within various in vivo animal models, with exercise leading to sustained calcium leakage, increasing both exercise tolerance and markers of mitochondrial biogenesis [208,209,210].

### 5.8. p38 Mitogen-Activated Protein Kinase (p38 MAPK)

To counter tissue damage caused by exercise, the body releases cytokines into the bloodstream that cause inflammation to help protect from subsequent stress [211]. These molecules bind to extracellular receptors and stimulate pathways like the MAPK/ERK pathway—a kinase cascade that activates the p38 mitogen-activated protein kinase (p38 MAPK) in human skeletal muscle [212,213]. Additionally, there is evidence that ROS increase the activation of p38 MAPK, with a time and dose dependent relationship identified with in vitro skeletal muscle models [214,215]. The activity of p38 MAPK is also regulated by PERM1, with a knockdown of the later in mice severely decreasing p38 MAPK phosphorylation and its downstream targets [98]. The p38 MAPK upregulates mitochondrial biogenesis by phosphorylation and stabilisation of PGC-1α, further leading to its accumulation within the nucleus [213]. While the majority of p38 MAPK exists within the cytosol, upon the advent of stress through the aforementioned processes there is an increased accumulation of p38 MAPK within the nucleus of skeletal muscle [216,217]. Inside the nucleus p38 MAPK can activate both MEF2 and activating transcription factor 2 (ATF2), the latter being another regulator upstream on the promoter of the PGC-1α gene [218,219]. Evidence of increased p38 activity in human skeletal muscle has been demonstrated by immunoblotting, with an increase in phosphorylated p38 MAPK nuclear content but no change in total whole-muscle protein content after exercise [192,220]. However, no direct observation of an increase in nuclear p38 MAPK protein content following exercise in human skeletal muscle has been published.

### 5.9. Hypoxia-Inducible Factor 1 (HIF1)

During exercise the level of oxygen within skeletal muscle decreases as its consumption is increased to fuel contraction, triggering various metabolic changes that influence mitochondrial biogenesis [221,222,223]. Decreases in cellular oxygen, termed hypoxia, are sensed through hypoxia-inducible factor 1-alpha (HIF1α)—a subunit of the transcription factor HIF1 [224]. Unlike other tissue, skeletal muscle maintains a comparatively high expression of HIF1α, even under normal oxygen conditions (normoxia) [224]. Under normoxia certain amino acid residues in HIF1α are oxidised, allowing binding of Von Hippel–Lindau tumour suppressor (pVHL) and flagging the former for degradation by ubiquitin-mediated proteolysis [225,226]. Although exercise in normal atmospheric conditions only marginally decreases the partial pressure of oxygen (pO_2_) in human skeletal muscle, HIF1α is still increasingly stabilised immediately after a single session of exercise [223]. Under hypoxia, stabilised HIF1α can localise to the nucleus of skeletal muscle through a C-terminal NLS and bind with HIF1β to gene promoters and other transcription factors [70,223,227,228,229].

Recently, HIF1α has been identified as accumulating in the mitochondria in human in vitro and rat ex vivo cell models during hypoxia [230,231,232]. Although seemingly inaccessible to the inner mitochondrial matrix, it is speculated to lower mtDNA transcription and prevent oxidative phosphorylation [230,231]. Additionally, HIF1α was found to localise to the OMM and displace mitochondrial fission protein dynamin-related protein-1 (Drp1); this reduces the proportion of proteolytically cleaved fusion dynamin like 120 kDa protein (OPA1) and inhibits membrane fission, with both fission and fusion after exercise critical to maintaining a healthy population of mitochondria [232]. However, whether exercise can induce HIF1α localisation to the mitochondria in human skeletal muscle is highly speculative and warrants significantly more research before any conclusion can be drawn.

### 5.10. Mammalian Target of Rapamycin (mTOR)

The protein synthesis regulator, mammalian target of rapamycin (mTOR), is a key metabolic sensor that responds to various stimuli occurring during exercise. Stimuli that affect mTOR complexes include the extracellular binding of insulin and cytokines to external receptors, amino acid availability, and fluxes of intracellular ATP levels [233,234]. To exert its function in humans, mTOR binds to various co-factors to form two separate complexes in the cytosol termed mTORC1 and mTORC2 [235]. During exercise downstream modifications are made to the proteins that regulate mTOR, including the degradation of the DEP domain-containing mTOR-interacting protein (DEPTOR), leading to the activation of mTORC1 [236,237,238,239,240,241,242].

Primarily, mTORC1 has been implicated in the driving of myofibrillar and mitochondrial protein synthesis via inhibition of translation inhibitor 4E-BP1 and translation activator p70-S6 Kinase 1 (S6K1) through phosphorylation in the cytosol [243]. Upon activation, mTORC1 also localises to the nucleus where it can form a complex with PGC-1α and yin-yang 1 (YY1) that binds upstream of numerous OXPHOS genes within in vitro mouse skeletal muscle cells [242,244]. In an in vitro mouse skeletal muscle model, electrically stimulated contraction caused an increase in the phosphorylation and activation of the mTORC1 complex in the nucleus two hours post contraction [245]. However, this study utilised subcellular fractionation and hence contamination from other compartments may have led to this finding. While it is clear mTOR plays a role in regulating mitochondrial biogenesis, further research is needed to understand the importance of its localisation post exercise.

### 5.11. Small Open Reading Frame Encoded Peptides (SEPs)

Outside of traditional transcription factors, there are likely other groups of proteins that regulate mitochondrial biogenesis following exercise. This may include small open reading frame encoded peptides (SEPs)—proteins consisting of approximately 20 to 80 amino acids that typically reside in stretches of DNA consisting of less than 300 nucleotides and which were previously thought to be non-coding [246]. As a result of improvements to non-targeted techniques, such as RNA sequencing and mass spectrometry, detection of SEPs is increasing exponentially [246]. Recent research has indicated mitochondria are enriched for SEPs, with experimental validation of not just SEPs encoded within mtDNA but also SEPs encoded within the nucleus that subsequently localise to the mitochondria [246]. Due to the process of importing large proteins into the mitochondria being energetically costly, it is hypothesised that SEPs may be favoured as a means of signal transduction between organelles [246].

The mitochondrial open reading frame of the 12S ribosomal RNA type-c (MOTS-c) peptide is a mitochondrial-encoded SEP that significantly increases post exercise in human skeletal muscle [247]. Nuclear accumulation of MOTS-c in human skeletal muscle has been identified in the 4-h post exercise, leading to an upregulation of gene expression for PGC-1α and mitochondrial proteins [247,248,249]. The mitochondrial encoded SEP humanin has also been implicated in regulating mitochondrial biogenesis following exercise in skeletal muscle through increasing mRNA expression of PGC-1α and its downstream targets, such as TFAM and NRF1 [250,251]. Evidence for an accumulation of humanin directly after exercise in the nucleus of human skeletal muscle has not been shown but is suspected to occur [250]. Because of recent discoveries such as MOTS-c and humanin, it is likely that there are other undetected SEPs encoded in both the nucleus and mitochondria that move between subcellular compartments to regulate mitochondrial biogenesis.

## 6. Future Directions

Although the study of exercise-induced mitochondrial biogenesis has advanced substantially over the past 60 years, many questions remain unanswered. Central to an understanding of mitochondrial biogenesis is identifying transcription factors that move between compartments to regulate the expression of mitochondrial proteins.

Understanding how different intensities of exercise, such as high-intensity interval exercise (HIIE), sprint interval exercise (SIE), and moderate-intensity continuous exercise (MICE), affect transcription factor localisation may enable researchers to identify how intensity-specific mitochondrial adaptations occur [252]. In a similar vein, how the exercise-induced movement of transcription factors differs between fibre types, the classification of myocytes based on structural and metabolic characteristics, is a key area of interest. The advancement of single-fibre proteomics has already begun to unravel some of these differences; however, identifying subcellular movement in single fibres remains a challenge due to the small amount of protein available for analysis [253,254]. Due to the reliance on using whole muscle or non-specific mitochondrial isolation techniques to analyse changes following exercise, differences in the movement of transcription factors in and out of IMF and SS mitochondria is under researched. Due to the varying metabolism between IMF and SS mitochondria, it is prudent that new subcellular fractionation techniques are capable of separating the two populations to identify differences in transcription factor movement. The influence of biological sex on exercise-induced mitochondrial biogenesis remains under researched as well, with some differences in transcript abundance already reported post resistance exercise [255]. This indicates that upstream differences in transcription factor movement following resistance exercise and potentially other forms of exercise likely exist between the sexes. It is imperative that future studies investigating exercise-induced movement of transcription factors use both men and women to account for any differences that may exist.

While PGC-1α mediated pathways remain the most investigated, those that operate in parallel, such as those involving TFEB, present new and promising areas to explore [161]. Transcription factors that are already known to heavily alter mitochondrial metabolism, such as PGC-1α, p53, and HIF1α, have not only been linked to gene expression in the nucleus, but also in the mitochondria [63,186,231]. It is therefore important to continue investigating these transcription factors, as well as others discovered in the future, by developing new techniques capable of identifying proteins from different subcellular compartments. Outside of those previously discussed, a technique that has yet to be applied to the subcellular proteome of skeletal muscle is BioID-based proximity labelling [256]. By attaching a biotin ligase to proteins with a known subcellular localisation or role in mitochondrial biogenesis, uncharacterised transcription factors that localise in the immediate intracellular environment can be biotinylated and detected using immunofluorescence. These biotinylated proteins can subsequently be isolated and identified using currently available mass spectrometry-based proteomic techniques.

To increase the depth of results, a protocol that can generate separated subcellular compartments and be analysed using not just one, but many techniques (including immunoblotting, RNA sequencing, and mass spectrometry-based proteomics) is needed. The integration of proteomics with metabolomics and transcriptomics, colloquially known as multi-omics, is increasingly being used to understand complex processes such as mitochondrial biogenesis [257]. By comparing multiple biological datasets, associations between different variables can be identified and used to build more complex models of exercise-induced mitochondrial biogenesis. This may highlight gaps within our understanding of mitochondrial adaptations via changes in metabolites, movement of transcription factors, and mRNA transcript abundance.

Due to their role in numerous cellular processes, mitochondria are critical for human health with various diseases characterised by sub-optimal mitochondrial function [258,259,260,261,262,263]. By understanding how the localisation of transcription factors influences mitochondrial adaptation, a greater tailoring of exercise intensity, duration, and frequency for achieving specific outcomes may be possible. The identification of novel transcription factors that move between subcellular compartments following exercise may also provide new targets for drug therapeutics that can improve mitochondrial function or recapitulate some of the effects of exercise [264,265].

## 7. Conclusions

Skeletal muscle is a tissue that is essential to human health, with the mitochondria a key component of its function. To adapt to exercise, the mitochondrion is capable of synthesising new protein components through the process of mitochondrial biogenesis. A single session of exercise can stimulate various transcription factors to move between the different subcellular compartments in muscle to activate the transcription of mitochondrial proteins in both the nucleus and mitochondria. Due to the importance of localisation on transcription factor activity, further research using techniques capable of differentiating between proteins within the mitochondria, cytosol, and nucleus of muscle is warranted. Through the reappraisal of known transcription factors, it is likely novel regulators will also be discovered that have previously evaded detection.

## Figures and Tables

**Figure 1 ijms-23-01517-f001:**
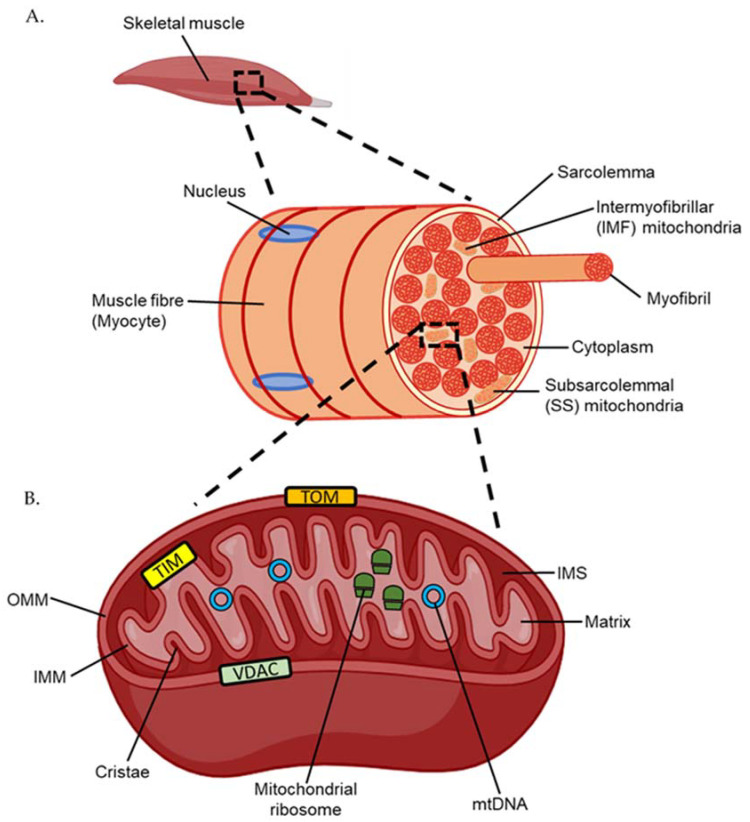
(**A**) Structure of a human muscle fibre or myocyte. (**B**) Structure of the mitochondria. TOM; translocase of the outer membrane. TIM; translocase of the inner membrane. VDAC; voltage-dependent anion channel. IMM; inner mitochondrial membrane. IMS; inner membrane space. OMM; outer mitochondrial membrane. mtDNA; mitochondrial DNA.

**Figure 2 ijms-23-01517-f002:**
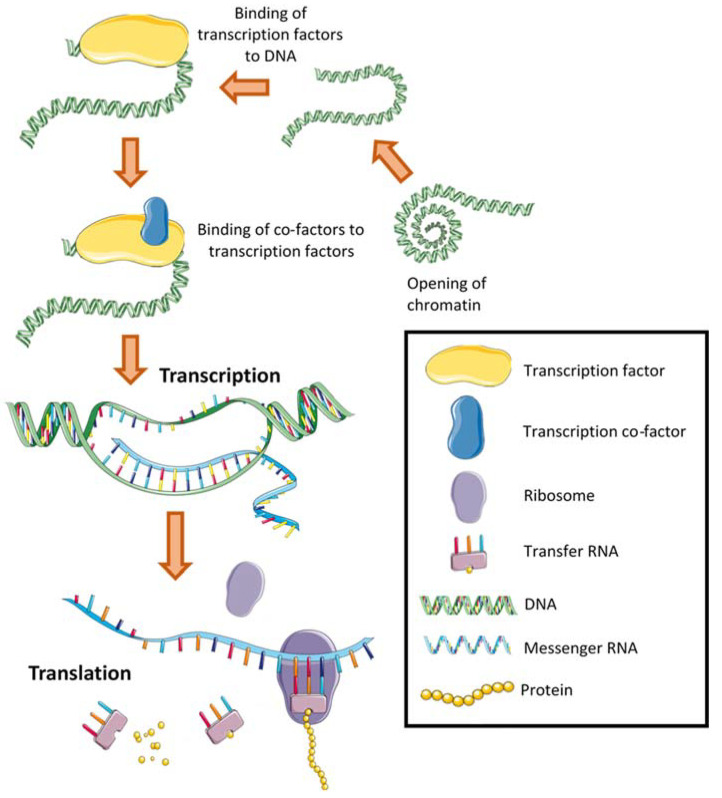
The regulation of gene transcription occurs through either several or all these processes. Each process may negatively or positively regulate the expression of a specific gene, influencing the resultant translation of messenger RNA into a protein. Histones and parts of the transcriptional machinery have not been shown for simplicity.

**Figure 3 ijms-23-01517-f003:**
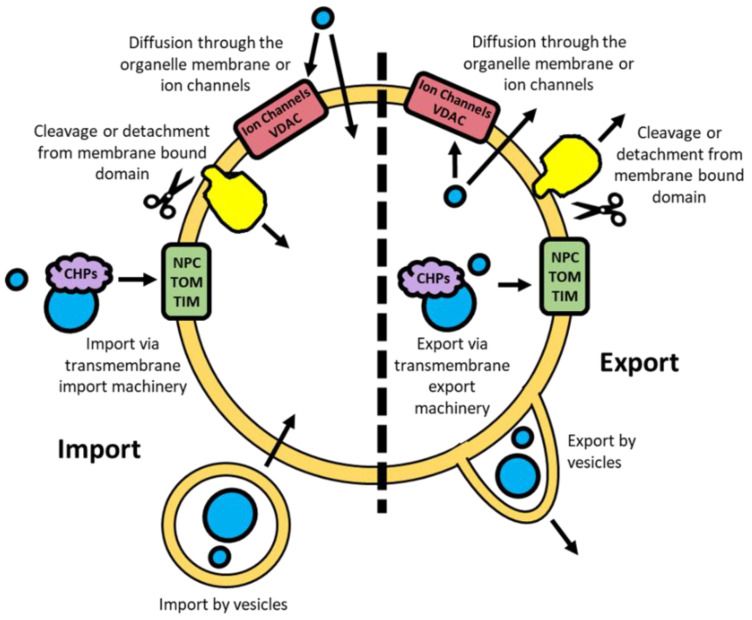
Import and export mechanisms for molecules passing through an intracellular nuclear or mitochondrial double membrane. NPC; nuclear pore complex. TOM; translocase of the outer membrane. TIM; translocase of the inner membrane. VDAC; voltage-dependent anion channel. CHPs; chaperones.

**Figure 4 ijms-23-01517-f004:**
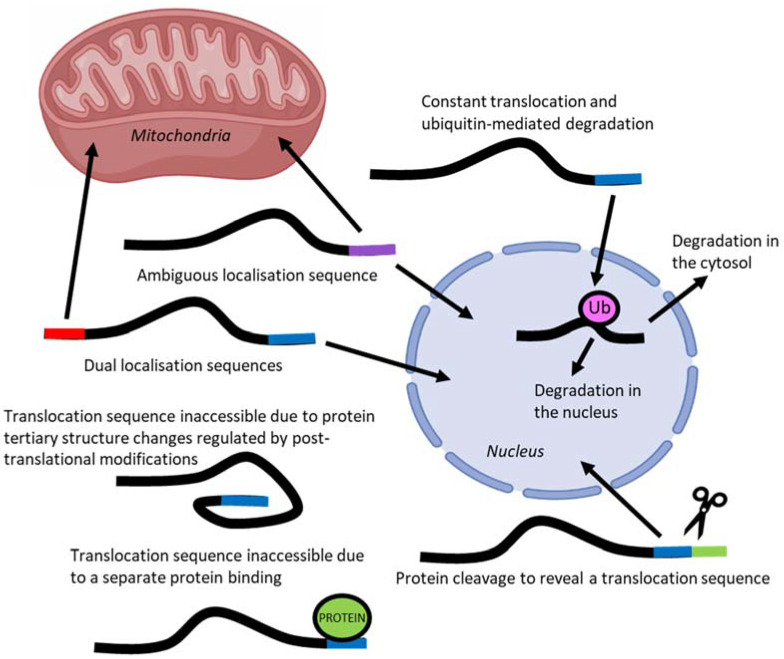
Regulation of protein translocation may occur through one or several of these mechanisms, each of which can additionally be controlled by post-translational modifications.

**Figure 5 ijms-23-01517-f005:**
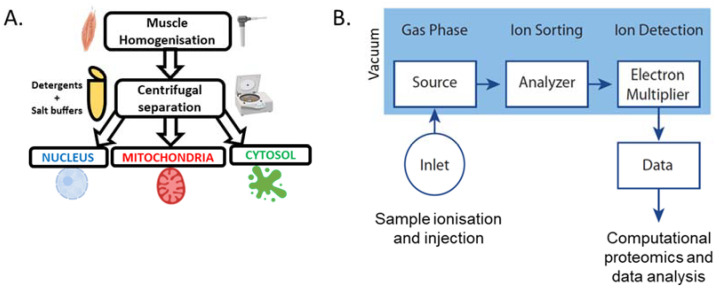
(**A**) Basic methodology for the subcellular fractionation of skeletal muscle [38,39,40]. (**B**) Basic overview of mass spectrometry workflow [45].

**Figure 6 ijms-23-01517-f006:**
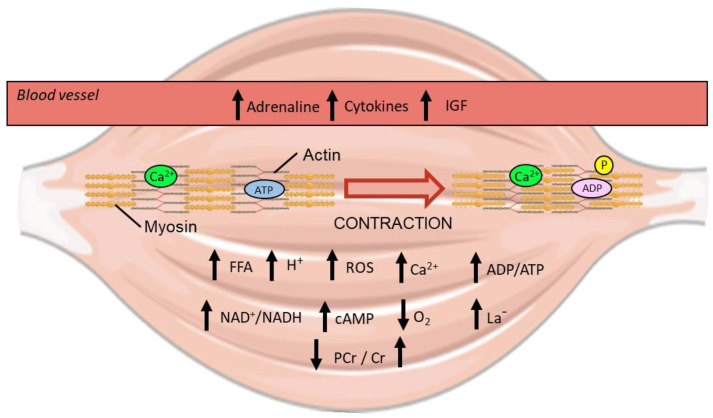
Visualisation of the homeostatic perturbations that occur during exercise within human skeletal muscle [65]. Up and down arrows indicate an increase or decrease in metabolite concentration respectively from exercise. IGF; insulin-like growth factor. ROS; reactive oxygen species. FFA; free fatty acid. PCr; phosphocreatine. Cr; creatine. Ca^2+^; calcium ion. P; phosphate. ADP; adenosine diphosphate. ATP; adenosine triphosphate. H^+^; hydrogen ion. NAD^+^; nicotinamide adenine dinucleotide. cAMP; cyclic adenosine monophosphate. La-; lactate.

**Figure 7 ijms-23-01517-f007:**
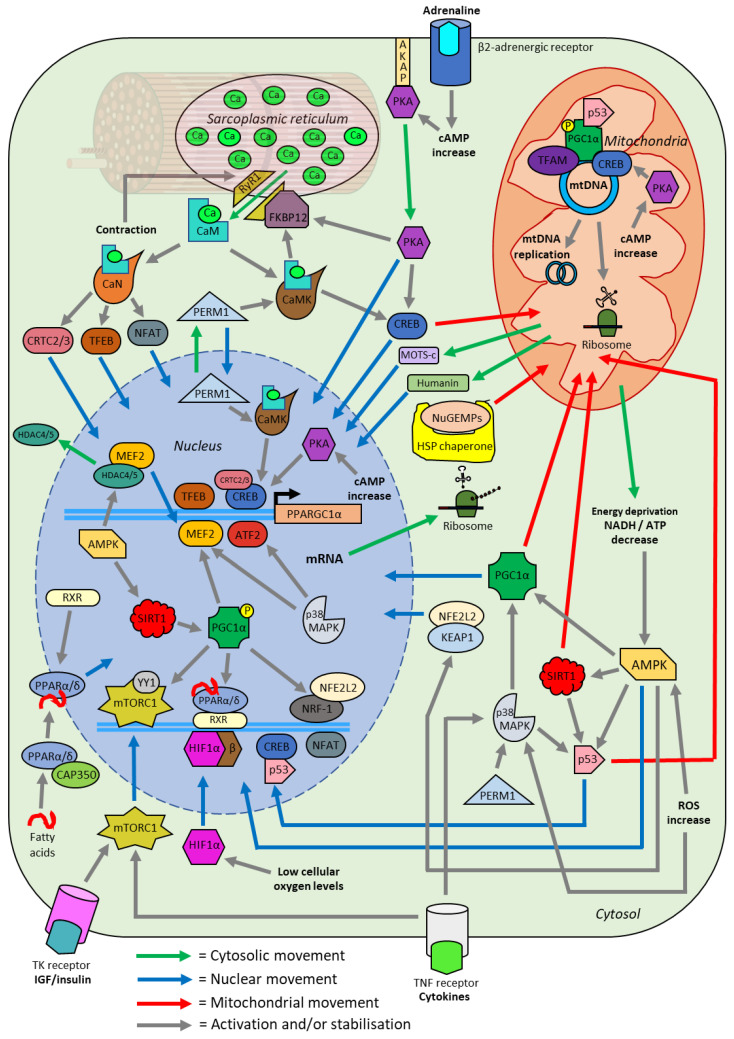
Schematic overview of the effects of a single session of exercise on transcription factor localisation. A single session of exercise causes various acute homeostatic perturbations within skeletal muscle, initiating signalling cascades (grey lines) in numerous pathways. These pathways facilitate the movement of transcription factors into the nucleus (blue line), cytosol (green line), and mitochondria (red line). These transcription factors lead to the upregulated expression of genes encoding mitochondrial proteins in a process termed mitochondrial biogenesis. All of the interactions depicted have either been experimentally verified or are likely to occur based on the current available evidence. ATP; adenosine triphosphate. NAD; nicotinamide adenine dinucleotide. cAMP; cyclic adenosine monophosphate. ROS; reactive oxygen species. CaM; calmodulin. CaN; calcineurin. CaMKII; Ca^2+^/calmodulin-dependent protein kinase II. CAP350; centrosome-associated protein 350. CRTC; CREB regulated transcription coactivator. SIRT1; NAD^+^-dependent deacetylase sirtuin-1. TFEB; transcription factor EB. AMPK; AMP-activated protein kinase. PGC1α; peroxisome proliferator-activated receptor gamma co-activator 1-a. IGF; insulin-like growth factor. TR; tyrosine kinase receptor. NuGEMPs; nuclear genes encoding mitochondrial proteins. mRNA; messenger RNA. mtDNA; mitochondria DNA. TFAM; mitochondrial transcription factor A. MEF2; myocyte-specific enhancer factor 2. PKA; protein kinase A. FKBP12; FK506-binding protein 12. RyR1; ryanodine receptor 1. PERM1; peroxisome proliferator-activated receptor gamma coactivator 1- and estrogen-related receptor-induced regulator in muscle 1. MOTS-c; mitochondrial open reading frame of the 12S ribosomal RNA type-c. NFAT; nuclear factor of activated T-cells. CREB; cyclic adenosine monophosphate response element binding. HSP; heat shock protein. HDAC; Histone deacetylase. ATF2; activating transcription factor 2. MAPK; mitogen-activated protein kinase. NFE2L2; nuclear Factor, erythroid 2-like 2. KEAP1; kelch ECH associated protein 1. PPAR; peroxisome proliferator-activated receptor. YY1; yin-yang 1. mTORC; mammalian target of rapamycin complex. HIF1α; hypoxia-inducible factor 1-alpha. NRF; nuclear respiratory factor. RXR; retinoid X receptor. TNF; tumour necrosis factor.

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
