# Peer review of "Transcription Factor Movement and Exercise-Induced Mitochondrial Biogenesis in Human Skeletal Muscle: Current Knowledge and Future Perspectives"

_ijms, 2022, doi:10.3390/ijms23031517_

Round 1

Reviewer 1 Report

The authors have prepared a thorough and comprehensive review of transcription factor localization/regulation that promotes mitochondrial biogenesis in skeletal muscle in response to exercise. Such a comprehensive compilation of studies about these transcription factors should be a useful addition to the field. I appreciated the combination of helpful reviews, classic papers, and brand new papers cited in support of those goals. The review is  written with clarity and enjoyable-to-read language. I have only minor comments for the authors to consider:

Regarding figures: 

Figure 2: Consider adding a line to the figure legend (lines 96-98) about histones and transcriptional machinery not being shown for simplicity.

Figure 4 is cited in line 131 right after mentioning chaperones, but there are no molecular chaperones illustrated in the figure. Consider including a broad summary sentence about Figure 4 at the beginning of Section 3.2 and putting the figure citation there.

Figure 5D: The description in the figure legend is very helpful, but I was confused by the main text in lines 176-178. Instead of "these fractions", perhaps use "the nuclear fraction" because, if my understanding is correct, only the nuclear fraction was subjected to proteomics analysis in 5D.

Figure 5D, 2nd question: For the % of genes detected, approximately how many correspond to hypothesized vs experimentally validated protein localization GO terms? Are the results better matched by the experimentally validated localizations? Would the authors like to comment on that in the text if it clarifies the point about the challenge with in silico predictions of localization?

Figure 6: Would it be useful to specify in the figure legend that the up arrows mean increased in exercise?

Figure 7: This figure has a lot going on, but the color coding and shapes designed by the authors are very helpful. I have two minor questions: 1) Underneath the mito, "Energy deprivation/ NAD+/ATP decrease": Do you mean that NADH and ATP decrease? (NAD+ to NADH ratio increases with exercise) 2) Was the connection between PERM1 and p38/MAPK referenced in the text? I may have missed it.

Additional minor points from the text:

Line 40-41: Consider introducing VDAC here as a pore through which small molecules can enter the IMS. VDAC appears in Figure 1 and is not mentioned in the text until p.5

Line 77: Is this the intended reference (#13)?

Line 81-82: Instead of "influencing" perhaps specify the direction: "...histone compaction heavily decreasing rates of transcription."

Line 105: "various metabolites": Perhaps include lipids (ligands for PPARs) here among discussion of ligands which can cross membranes

Line 120: "passively diffuse into or out of the IMS..."

Line 143-150: Please clarify that Pink1/Parkin are not transcription factors.

Line 152: Perhaps elaborate on why reference 37 was cited here? Is it because p53 nuclear localization was changed without a decrease in cytoplasmic levels?

Line 244: Perhaps elaborate on why reference 20 was cited here?

Line 274: Instead of "detachment of a phosphate", perhaps use "phosphate transfer"?

Line 281-282: Although the IMM was canonically thought to be impermeable to NAD+, consider mentioning the recent studies that have characterized the mammalian mitochondrial NAD+ transporter: https://pubmed.ncbi.nlm.nih.gov/32906142/

https://pubmed.ncbi.nlm.nih.gov/33087354/

Line 296: Is this intended reference (#70)? The authors of that paper mention that H2O2 diffuses well across membranes, unlike many other ROS species.

Line 300: Should insulin here be insulin-like growth factor instead (as in Figure 6)? If insulin was intended, please elaborate on why insulin would be increased by exercise.

Line 305: Consider specifying that additional cAMP is generated by soluble adenylate cyclase

Line 319-320: Please clarify that CAMKII is not a transcription factor but a regulator of these transcription factors.

Line 325: Please check the threonine phosphorylation site referenced. I think reference 85 and 88 show pT287

Line 339, 714: What is the full name of PERM1? Is it "peroxisome proliferator-activated receptor gamma coactivator 1- and estrogen-related receptor-induced regulator in muscle 1"?

Line 344: Should the reference about the increase in mtDNA content and CS activity in mice overexpressing Perm1 in skeletal muscle be this reference instead of 93: https://pubmed.ncbi.nlm.nih.gov/26481306/

Line 367: Is PGC-1a an intrinsically disordered protein, or is it just the activation domain that is intrinsically disordered? Consider adding a reference about this, perhaps: 

https://pubmed.ncbi.nlm.nih.gov/22049338/

Line 373-374: Perhaps include ERRs and RXRs here too (two other classes of transcription factors that interact with PGC-1a)?

Line 567: Was it intended to cite all the references from 180-186 here?

Line 567-568: Consider specifying how the phosphorylation status of p53 changes its localization and activity?

Line 590: Section numbering accidentally repeated: Should be 5.7 and all the subsequent sections need to be corrected too

Line 633, 658, 731 : Instead of "metabolomic" perhaps use the word "metabolic" here

Line 649-652: Consider briefly addressing whether exercise normally promotes mitochondrial fusion

Line 659-660: Is amino acid sensing another important function of mTOR in skeletal muscle upon exercise?

Line 668: Consider specifying that mTOR inhibits these translational components by phosphorylation?

Line 673: Is it worth mentioning that reference 242 only used subcellular fractionation (and the caveats associated with that)?

Line 746-749: Consider adding a brief discussion of proximity labeling approaches such as Bio-ID compartmentalization. Perhaps something like these papers: 

https://pubmed.ncbi.nlm.nih.gov/34425888/

https://pubmed.ncbi.nlm.nih.gov/34079125/

Thank you for preparing this well-written and comprehensive review! It was a pleasure to read!

Reviewer 2 Report

The authors realized a review on the exercise-induced transcription factor movement involved in mitochondrial biogenesis process. The review is well constructed, in the introduction, the authors described the muscle and the mitochondria. Next, an overview of the role of transcription factors in mitochondrial biogenesis is provided. The two following sections described the regulation and the assessment methods of transcription factor. The last and main part of the review, describe the exercise-induced movement of transcription factor in skeletal muscle.

I would like to congratulates the authors as the review is didactic and well-constructed. I have no major comment but only some minor suggestions:

  • L215-217: an example would be interesting for the reader.
  • L350-364: it would be interesting to add information about how exercise intensity or modality may influence CamKII phosphorylation (for example: PMID: 20308248, PMID: 26359238).
  • L415-416: as it is based only on one study in rat, it is may be too speculative?
  • L514-516: I think is no very intuitive for a naive reader on why inhibition of OXPHOS could be a stress similar to exercise?
  • L646-647: HIFa accumulation in the mitochondria is not presented in fig 6.
  • Future directions: is there an interest to evaluate specific exercise-induced transcription factor movement in IMF vs SS mitochondria.
  • Figure 4: I would suggest to add the post-translational modification possibility on the graph as depicted in the text.
  • Figure 5:
    • Legend: indicate to which cellular compartment correspond the letter C, M and N in the panel B
    • The panel D present data issue from panel B but the cytoskeletal fraction was not presented in panel B. It could be confusing for the reader
  • Figure 6:
    • I would suggest to present abbreviation used in the figure in alphabetical order.
    • For me ADP, TOM and TR are note present on the figure and TK, IGF, CAP350 and AMPK are not described in the legend.
